METHODS

# Generating correlated data for omics simulation

**Jianing Yang**[1,2], **Gregory R. Grant**[1,3], **Thomas G. Brooks** [1]*

**1** Institute for Translational Medicine and Therapeutics, University of Pennsylvania, Philadelphia, Pennsylvania, United States of America, **2** Chronobiology and Sleep Institute, University of Pennsylvania, Philadelphia, Pennsylvania, United States of America, **3** Department of Genetics, University of Pennsylvania, Philadelphia, Pennsylvania, United States of America

* thobr@pennmedicine.upenn.edu

**Data availability statement:** Source code for all simulations and figures in this paper is available at github.com/itmat/ dependent_sim_paper/. Source code for the

## Abstract

Simulation of realistic omics data is a key input for benchmarking studies that help users obtain optimal computational pipelines. Omics data involves large numbers of measured features on each sample and these measures are generally correlated with each other. However, simulation too often ignores these correlations, perhaps due to computational and statistical hurdles of doing so. To alleviate this, we describe three approaches for generating omics-scale data with correlated measures which mimic real datasets. These approaches are all based on a Gaussian copula approach with a covariance matrix that decomposes into a diagonal part and a low-rank part. This decomposition allows for extremely efficient simulation, overcoming a hurdle for adoption of past methods. We use these approaches to demonstrate the importance of including correlation in two benchmarking applications. First, we show that variance of results from the popular DESeq2 method increases when dependence is included. Second, we demonstrate that CYCLOPS, a method for inferring circadian time of collection from transcriptomics, improves in performance when given gene-gene dependencies in some circumstances. We provide an R package, dependentsimr, that has efficient implementations of these methods and can generate dependent data with arbitrary marginal distributions, including discrete (binary, ordered categorical, Poisson, negative binomial), continuous (normal), or with an empirical distribution.

## Author summary

Modern techniques, including high-throughput sequencing, produce more data than ever before. To determine the optimal computational analysis methods for these data, benchmarks are often performed using simulated data. This simulated data needs to closely match realistic data in order for benchmarking to meaningful. An often neglected aspect of this is that measurements of different values are often correlated or dependent upon each other. Two possible reasons for this neglect could be that there is a lack of

`dependentsimr` package is available at github.com/tgbrooks/dependent_sim. All sequencing data used is available from the Gene Expression Omnibus (GEO) with accession numbers GSE151923, GSE81142, and GSE151565. Metabolomics data was obtained from MetaboAnalyst (https://api2.xialab.ca/api/download/metaboanalyst/plasma_nmr.csv).

**Funding:** JY received funding from the National Institute of Neurological Disorders and Stroke (5R01NS048471). TGB and GRG received funding support from the National Center for Advancing Translational Sciences Grant (5UL1TR000003). The funders had no role in this research, study design, data collection and analysis, the decision to publish, or the preparation of this manuscript.

**Competing interests:** The authors have declared that no competing interests exist.

guidelines on how to produce such data and also that methods to produce it are computationally expensive to run. We describe here three related methods that are both conceptually relatively simple and also highly computationally efficient. We demonstrated these on two applications which show how inclusion of these dependencies can affect the results of benchmarking. Lastly, we provide a software package to act as a reference implementations of these.

## Introduction

Omics data typically has far fewer samples than measurements per sample. This creates dual challenges in generating realistic simulated data for the purposes of benchmarking. First, there is not enough data to be able to estimate an accurate dependence structure (e.g., a correlation matrix). Second, generating omics-scale data with a specified correlation matrix is slow due to the typical $O(p^3)$ nature of these algorithms, where $p$ is the number of measurements per sample. Moreover, there is a lack of practical guidance on how to generate simulated data with realistic dependence. Simulators therefore often assume independence of the measurements, which does not reflect reality.

Here, we describe three related solutions which all offer good performance and ease-of-use even for large omics-scale problems. This expands off part of a larger discussion we published on best practices in omics benchmarking [1]. Our goal here is to show that generating correlated data does not have to be onerous and instead should be considered a baseline requirement when simulating data.

Using a Gaussian copula [2] approach (also referred to as NORTA, for "normal to anything" [3]), the marginal (univariate) distributions can have realistic forms. These solutions operate by taking a real, reference dataset and producing simulations that mimic it. However, a major challenge when simulating datasets is choice of the copula, which is a mathematical object that encodes the dependence between two or more variables. For Gaussian copulas, this is a choice of a covariance matrix. However, the sample covariance matrix of a reference dataset is a poor choice and we present three alternative choices.

There is a large body of literature on determining covariance matrices for various purposes. Shrinkage methods [4,5] improve estimates and guarantee that the estimated matrix is well-conditioned (and hence invertible). Sparse matrix methods [6] produce matrices where many of the pairwise correlations are zero. Similarly, other methods produce a sparse estimate of the inverse of the covariance matrix [7], which imparts pairwise independence of some variables conditional on all the others.

Copula approaches are flexible and have a long history [8] of use in fields such as economics [9]. More recently, copula approaches have been used for modelling or classification in multi-omics fields [10,11] and metagenomics [12]. Projects simulating omics with copulas include SPsimSeq [13] and scDesign2 [14]. Some methods have employed the "vine copula" approach in the context of single-cell or spatial omics simulation [15,16]. Vine copulas provide an alternative approach for choosing a copula which decomposes it into a set of 2-dimensional dependencies, however it comes at significant computational cost when done at the omics-scale. SeqNet uses instead a network approach for gene dependence [17].

Here, we describe three different strategies for determining covariance matrices, all of which enable efficient generation of simulated data due to following a specific form. Specifically, these all decompose into a diagonal part and a low-rank part, which speeds up generation of random vectors. First, the 'PCA' method uses principal component analysis (PCA) and picks the low-rank part such that the simulated data has the same variance in the top $k$

PCA components of the reference dataset. Second, the 'spiked Wishart' method fits $k$ components such that simulations with the same number of samples as the reference dataset will have, on average, the same PCA component variances as the reference. Unlike 'PCA', these variances are computed with respect to the simulated data's own PCA rather than using the PCA weights of the reference dataset. Third, the 'corpcor' method builds off the popular R library `corpcor` [4,5], which implements a James-Stein type shrinkage estimator for the covariance matrix as a linear interpolation of the sample covariance matrix and a diagonal matrix. No method exactly replicates the reference dataset's properties, indicating room for future research, but all improve upon the common approach of assuming independence. We implement this in an R package which supports normal, Poisson, DESeq2-based (negative binomial with sample-specific size factors [18]), and empirical (for ordinal data) marginal distributions.

We include two applications which demonstrate the effects of including dependence of measurements in simulated data when benchmarking computational pipelines. First, we simulate RNA-seq data with differential expression between two conditions. Using DESeq2 to determine the differentially expressed genes, we found that dependence had little impact on the accuracy of reported $p$-values but increased the variance of those estimates. Second, we simulated a time series of RNA-seq data points and used the CYCLOPS method [19] to infer collection time from the RNA-seq data, without time labels. Depending upon settings used, the performance of CYCLOPS depends substantially on the dependence structure of the data, and surprisingly showed improved performance when given data with dependence.

## Results

We start with a reference dataset $X$ given by a $p \times n$ data matrix of $p$ features measured in each of $n$ independent samples, which could be any real dataset of interest. Our goal is to be able to generate simulated datasets that look the same as this reference dataset, on average and when $n$ samples are simulated. To do this, we need to capture correlations between the $p$ features, which could represent gene expressions, protein abundances, or other measured values. We refer to these features as genes for simplicity.

Our methods are based off the well-known Gaussian copula method, which uses a multivariate normal distribution to capture aspects of the gene-gene dependence, without necessarily assuming that the data is normally distributed. See Fig 1A for a high-level overview. This multivariate normal distribution is parameterized by a $p \times p$ covariance matrix, $\Sigma$. In omics, choosing the covariance matrix $\Sigma_{sim}$ to use during simulation presents a challenge since $n < p$. The most obvious choice is to use the sample covariance matrix $\widehat{\Sigma}_{ref}$ of the reference dataset (possibly after some transformation of the data). However, using $\widehat{\Sigma}_{ref}$ is a poor choice for $\Sigma_{sim}$. Indeed, we consider a choice of $\Sigma_{sim}$ to be successful if datasets of $n$ samples simulated using $\Sigma_{sim}$ have sample covariance matrices $\widehat{\Sigma}_{sim}$ that are similar to $\widehat{\Sigma}_{ref}$. Counterintuitively, data simulated using $\Sigma_{sim} := \widehat{\Sigma}_{ref}$ have sample covariance matrices that systematically and substantially differ from $\widehat{\Sigma}_{ref}$, see Fig 1B.

One reason this occurs is that $\widehat{\Sigma}_{ref}$ is low rank, specifically at most rank $n-1$, which is much less than $p$ in typical omics studies. This means that any data generated using $\Sigma_{sim} = \widehat{\Sigma}_{ref}$ will lie on an $n-1$-dimensional hyperplane. Intuitively, this means that the entire probability distribution of potentially $p$ dimensional data has been squished into a $n-1$ dimensional space, generating much higher gene-gene correlations than in the reference dataset. In contrast, we know that omics contains at least some level of random noise that is essentially independent from one gene to the next. For example, RNA-seq involves a random sampling of fragments from the true sample of RNA molecules and other omics modes involve various background

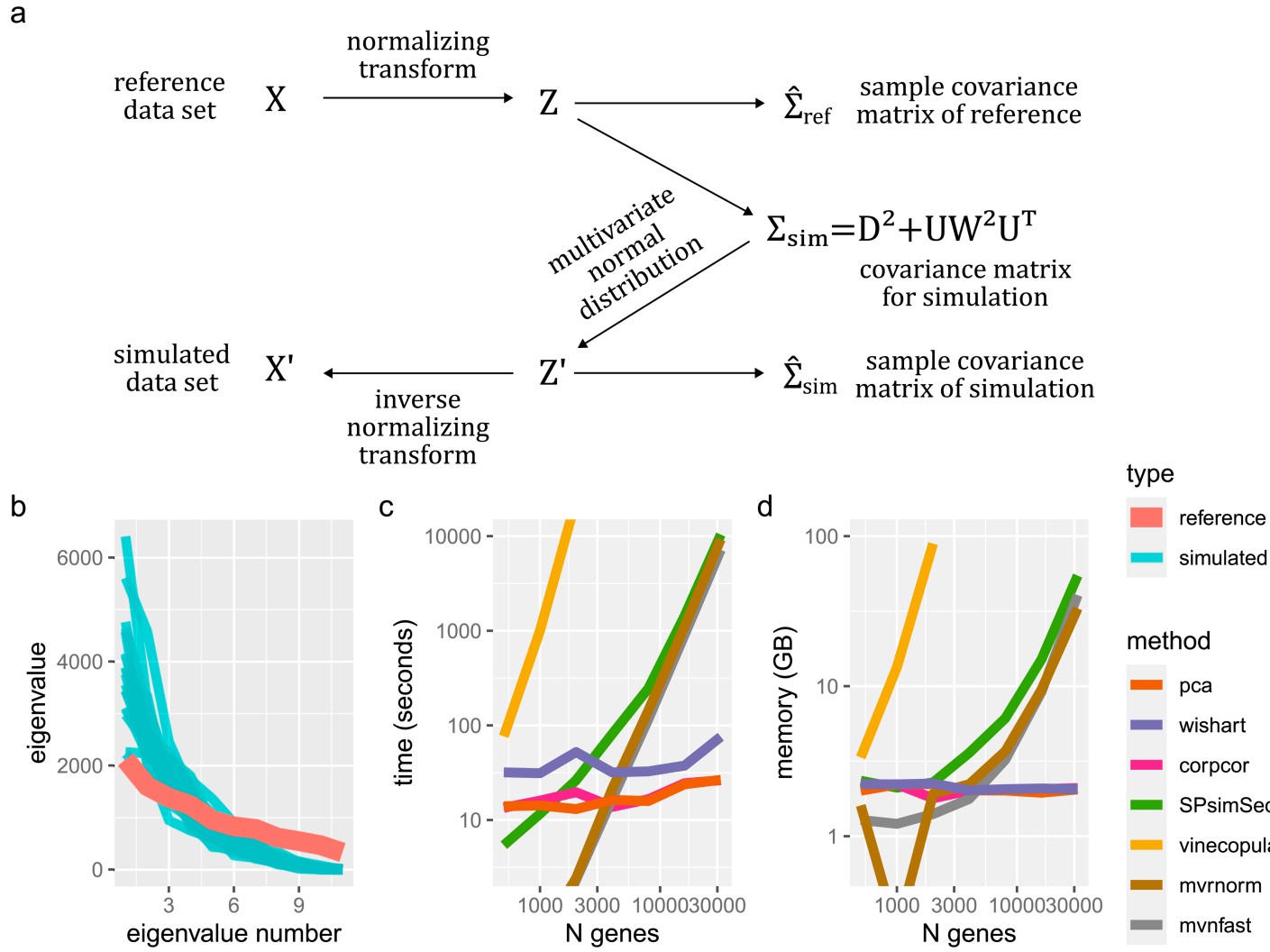

**Fig 1. (a) Schematic of our Gaussian copula-based approach.** The goal is to pick $\Sigma_{sim}$ so that $\widehat{\Sigma}_{sim} \approx \widehat{\Sigma}_{ref}$ while using a special, highly efficient form. (b) Eigenvalues of the sample covariance matrix $\widehat{\Sigma}_{ref}$ of reference data compared to $\widehat{\Sigma}_{sim}$ from simulated data using $\Sigma_{sim} := \widehat{\Sigma}_{ref}$. 100 genes and 12 samples were used for both reference and the 20 simulated datasets. Both reference and simulated data follow a multivariate normal distribution. Eigenvalues are numbered from largest to smallest. (c-d) Comparison of (c) run time and (d) memory usage for our methods (pca, wishart, corpcor), SPsimSeq, mvrnorm (from the MASS package), the mvnfast package, and vine copula (implemented by the rvinecopulib R package). 100 samples were generated with a varying number of genes. A mouse cortex dataset GSE151564 was randomly subsetted to have the required number of genes to be used as the reference data set. Vine copula method was only run up to 2000 genes due to the memory use required at higher gene counts. All methods used a single thread.

noise effects. We therefore need to increase the amount of independence beyond what $\widehat{\Sigma}_{ref}$ has, a process we refer to as "injecting" independence.

We present three methods - to be called PCA, spiked Wishart, and corpcor - for making the choice of $\Sigma_{sim}$ All three take the form

$$\Sigma_{sim} = D^2 + UW^2U^T$$

where $D$ is a diagonal matrix capturing the injected independence and $UW^2U^T$ is a low-rank correlation matrix derived from $\widehat{\Sigma}_{ref}$. An additional advantage of this special form is that data

may be generated very efficiently (Fig 1B and 1C) compared to using arbitrary $\Sigma_{sim}$ which typically require a Cholesky or eigenvalue decomposition. Specifically, random samples are generated using the fact that if $u, v$ are independent vectors with iid standard normal entries, then $Dv + UWu$ has covariance matrix equal to $D^2 + UW^2U^T$, see Methods.

## Comparison to real data

To compare the three simulation methods with a real dataset, we chose as a reference dataset 12 mouse cortex RNA-seq samples from accession GSE151923 [20]. We then simulated data mimicking this reference using all three simulation methods (PCA, spiked Wishart, and corpcor) as well as a simulation with independent genes. All data were simulated with negative binomial distributions fit by DESeq2 on unnormalized read counts. We repeated the simulations, each of 12 samples, a total of 8 times to estimate variance. For the PCA method, we used $k = 2$ dimensions and for the spiked Wishart, $k = 11$. The coprcor method always uses the full data matrix, analogous to $k = 11$. Note that PCA method must use a rank $k < 11$ in order to generate full-rank data, see Methods, so these parameters are not directly comparable across methods.

For comparison we also ran SPsimSeq, another RNA-seq simulator for both bulk and single-cell that uses a Gaussian copula-based model of dependence, see S1 Text for details.

Simulated data captures the genes' mean and variance accurately (Fig 2A and 2B). Next, we compared to the real dataset when projected onto the top two principal components of the real dataset (Fig 2C). The simulations with dependence are distributed around the entire space like the real data, but the independent simulations have unrealistically low variance in these components, clustering tightly around the origin.

Then, we computed the gene-gene correlation on pairs of high-expressed genes (at least 100 mean reads in the real dataset). The simulation with independence showed the least levels of gene-gene correlations (Fig 2D). However, the PCA method overshot the reference dataset and the spiked Wishart and corpcor methods modestly improved upon the independent simulation. The SPsimSeq simulator performed similarly to the PCA method.

Lastly, we compared the variances of principal components on each dataset (Fig 2E). These were computed separately for each dataset, unlike (Fig 2C) which used the reference dataset's PCA weights for all datasets. The independent data has much lower variance than the real dataset in the top four principal components. The spiked Wishart method comes closest to the real dataset, as it optimizes for fitting these values. Surprisingly, the corpcor method performs only somewhat better than the independent method. The PCA method puts a large amount of variance into the first two components (due to using $k = 2$) and then undershoots the other components. Like PCA, the SPsimSeq simulator, overshot the first PCA component but did a better job matching lower PCA coordinates.

We further demonstrated our methods on metabolomics data, where we use normal marginal distributions and with just 42 features measured, see S1 Fig.

## DESeq2 application

We benchmarked DESeq2 [18], a popular differential expression analysis tool, using datasets simulated with dependence and ones simulated without dependence to compare its performance on both. DESeq2 presents an interesting case because several aspects of it assume independence of genes and so may be adversely affected by gene-gene dependence. First, the independent filtering step [21] assumes independence but has been reported to be robust to typical gene-gene dependence. Relatedly, the false discovery rate (FDR) [22] allows only

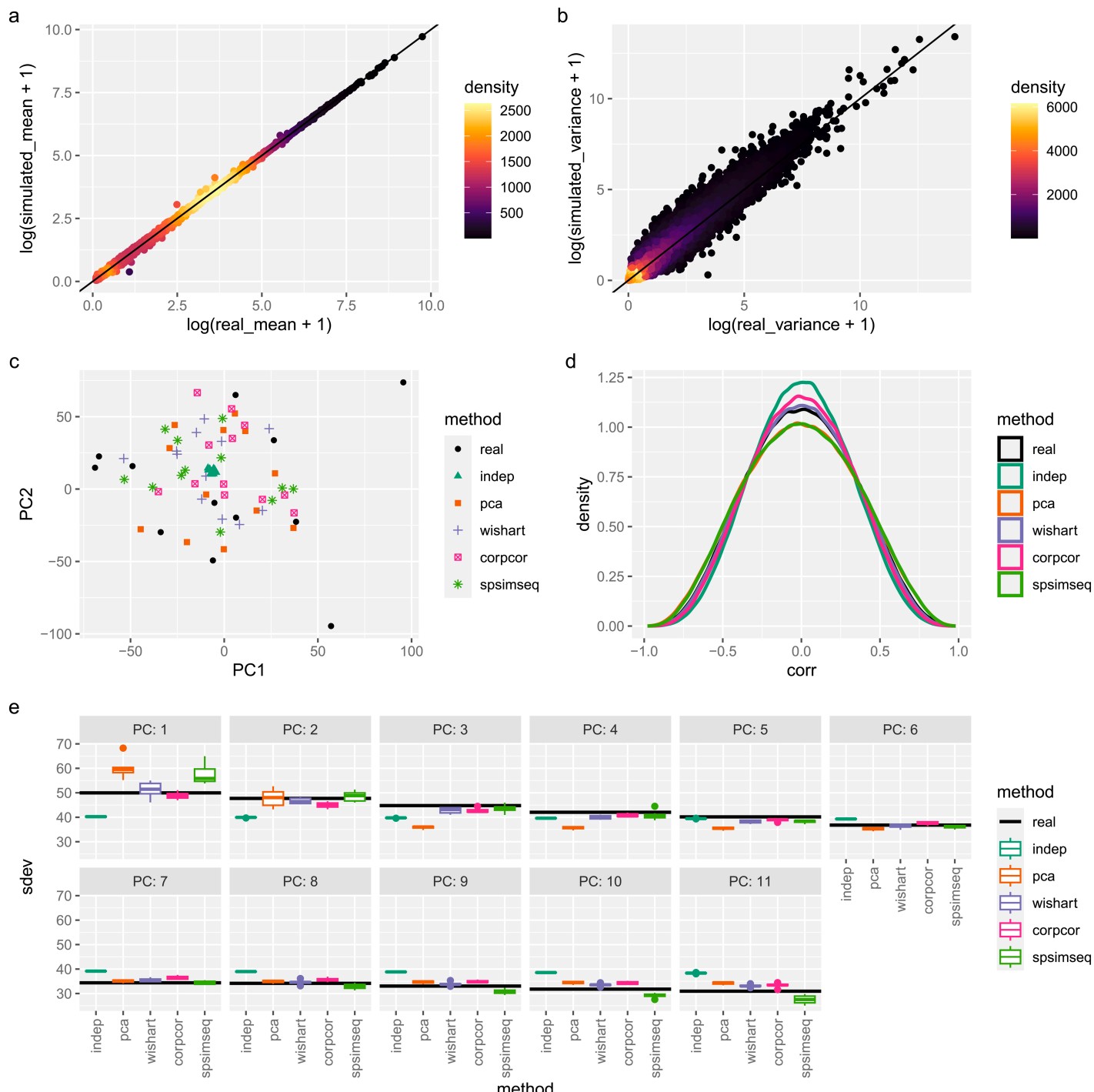

**Fig 2. Comparison to real reference data run on a mouse cortex dataset from GSE151923.** (a–b) Comparison of gene (a) mean expression and (b) variance, log-scaled in real and PCA simulated data. The line of equality is marked in black. Points are colored according to the density of points in their region. (c) Quantile-quantile plot comparing correlation values of gene pairs from real data and simulated data (both with and without dependence). Genes with at least 100 reads were used. Values on the diagonal line indicate a match between the simulated and real datasets. (d) Projections onto the top two principal components of the real dataset for both real and simulated data. All 8 simulations (96 samples for each simulation) shown. (e) Principal component analysis was performed on all datasets and the variance captured by the top components is shown. Unlike (d), these components were fit from each dataset considered separately instead of reusing the weights from the real data.

certain forms of dependence [23]. Lastly, DESeq2's empirical Bayes steps could possibly be affected by gene dependence.

We used a fly whole body RNA-seq dataset GSE81142, see S2 Fig. To select a "control" dataset samples were selected if they met the following criteria: male flies; without treatment; and after at least 2 hours of feeding. Unnormalized read counts were used as the reference dataset and all simulations used the DESeq2-based negative binomial model. We then randomly selected 5% of the genes to be differentially expressed, with absolute $\log_2$ fold change uniformly distributed between 0.2 and 2.0, either up or down regulated chosen randomly, and simulated 5 "experimental" samples. This was repeated 20 times for each of four dependence methods (independent, PCA, Wishart, and corpcor).

Finally, we ran DESeq2 on each simulated 5 vs 5 experiment and compared the output FDR with the true percentages of genes that are differential expressed (Fig 3A–3D). We observed that DESeq2 is anti-conservative on all datasets, with similar mean true FDRs for each estimated FDR cutoff. While the mean behavior was consistent whether or not the simulation included gene-gene dependence, simulations with dependence showed a substantially greater variance in the performance of DESeq2.

To demonstrate the application of our simulation method for another organism, we also simulated datasets using the mouse cortex dataset GSE151923 [20] and selected samples from male mice, as in Fig 2. We then simulated differential expression experiments as above and observed a similar result (Fig 3E and 3F) as for the fly whole body datasets.

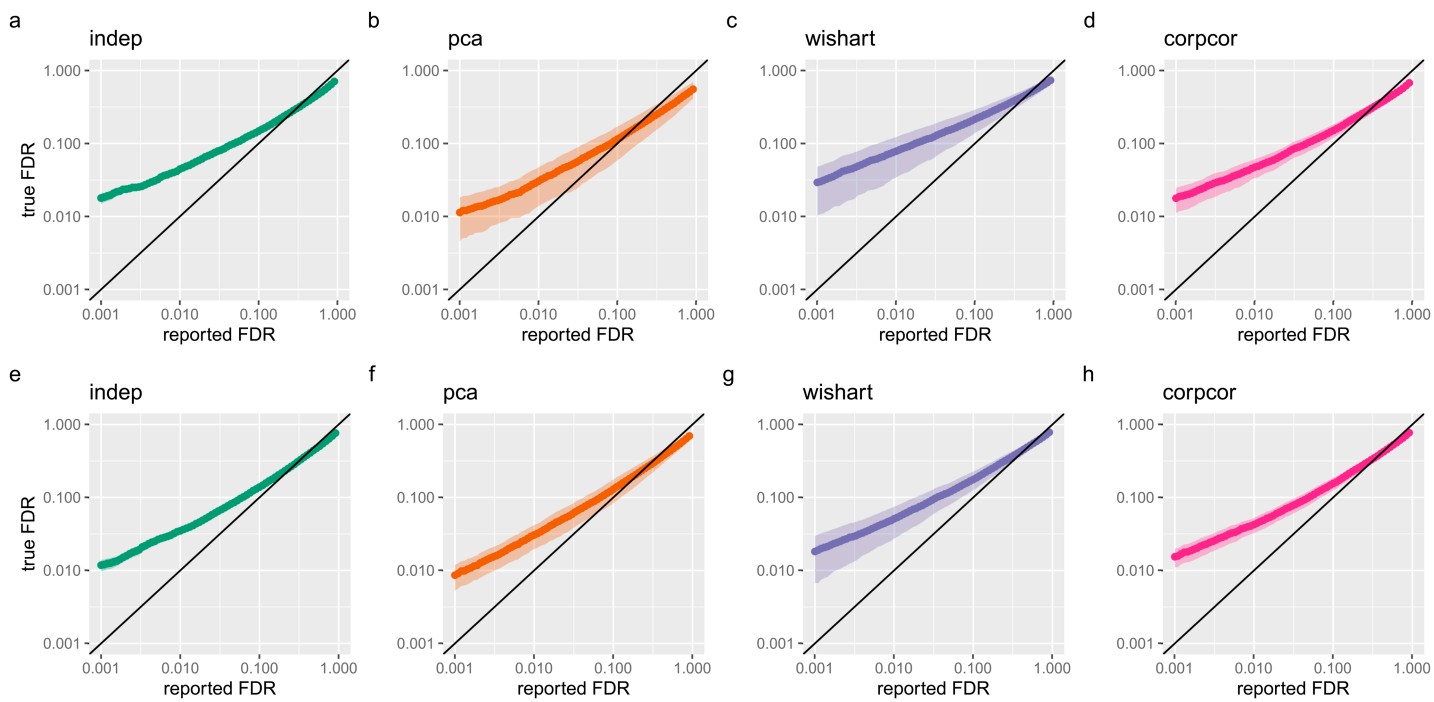

**Fig 3. Performance of DESeq2 on simulated datasets.** (a–d) Comparison of true false discovery proportions and DESeq2 reported False Discovery Rates, plotted on a log scale, for datasets simulated from the fly whole body dataset (GSE81142), (a) without dependence, (b) using PCA, (c) using Wishart and (d) using corpcor. Diagonal line represents perfect estimation of FDR. (e–h) Comparison of true false discovery proportions and DESeq2 reported FDR for datasets simulated from the mouse cortex dataset (GSE151923), (e) without dependence, (f) using PCA, (g) using Wishart and (h) using corpcor.

## CYCLOPS application

We next used our simulation method to benchmark CYCLOPS [19], which infers relative times for a set of unlabeled samples using an autoencoder to identify circular structures. We chose a mouse cortex time series dataset GSE151565, see S3 Fig, which contains a total of 77 samples every 3 hours, for 36 hours. As before, unnormalized read counts were used as the reference dataset and all simulations used the DESeq2-based negative binomial model. We computed the dependence structure of the genes as well as the variances of marginal distributions using the 12 time point 0 samples. We computed the means of gene expressions at each time point. We then used these to simulate 20 time series datasets for each of the four simulation methods: independent, PCA, Wishart, and corpcor.

We note that, unlike many timeseries methods that consider correlation across time points, the dependence here is between genes within the same sample and generated samples are independent, conditional upon the time point. This reflects scenarios where CYCLOPS is used, for example, post-mortem biopsy samples of unrelated individuals collected at different times [24].

We ran CYCLOPS on each dataset with a list of cyclic mouse genes (from [25], JTK p-value <0.05), which yielded an estimated relative time for each sample. We evaluated CYCLOPS' performance compared to true circadian time using the circular correlation [26], defined as follows:

$$\rho = \frac{\sum_{1 \le i < j \le n} \sin(X_i - X_j) \sin(Y_i - Y_j)}{(\sum_{1 \le i < j \le n} \sin(X_i - X_j)^2)^{1/2} (\sum_{1 \le i < j \le n} \sin(Y_i - Y_j)^2)^{1/2}},$$

where $n$ is the number of samples, $X_i$ and $Y_i$ are the true time and CYCLOPS-estimated time, respectively, for the $i$-th sample. $\rho$ has value between −1 and 1, and a $|\rho|$ close to 1 indicates accurate predictions by CYCLOPS.

By default, CYCLOPS performs dimension reduction so that each dimension (called an "eigengene") contains at least 3% of the total variance. We found that CYCLOPS accuracy depended significantly on this parameter, with the default producing good accuracy across all simulations. However, when dropping CYCLOPS to require just 2% variance in each eigengene, we found that its accuracy depends significantly on the dependence structure of the simulated time series data (Fig 4A). At that setting, CYCLOPS accuracy is much higher in the PCA method and moderately improved in Wishart method, compared to the 'independent' method. This difference is likely driven by the difference in the number of eigengenes used (Fig 4B), which is a measure of how much dependence is present in the dataset. Notably, the number of eigengenes and accuracy tracked both with each other and with the amount of gene-gene correlation (see S3D Fig) across the methods. This demonstrates that the correlation structure of the transcriptome can have a major impact on accuracy.

## Methods

Here, we provide an introduction to the Gaussian copula approach before describing our three methods for choosing $\Sigma_{\text{sim}}$.

### Multivariate normal distribution

We first discuss the simplest case, where our dataset is multivariate normally distributed. The distribution $N(\mu, \Sigma)$ is the multivariate normal distribution with mean vector $\mu$ and covariance matrix $\Sigma$. Here $\mu$ is a $p \times 1$ column vector and $\Sigma$ is a $p \times p$ matrix. In order for this to

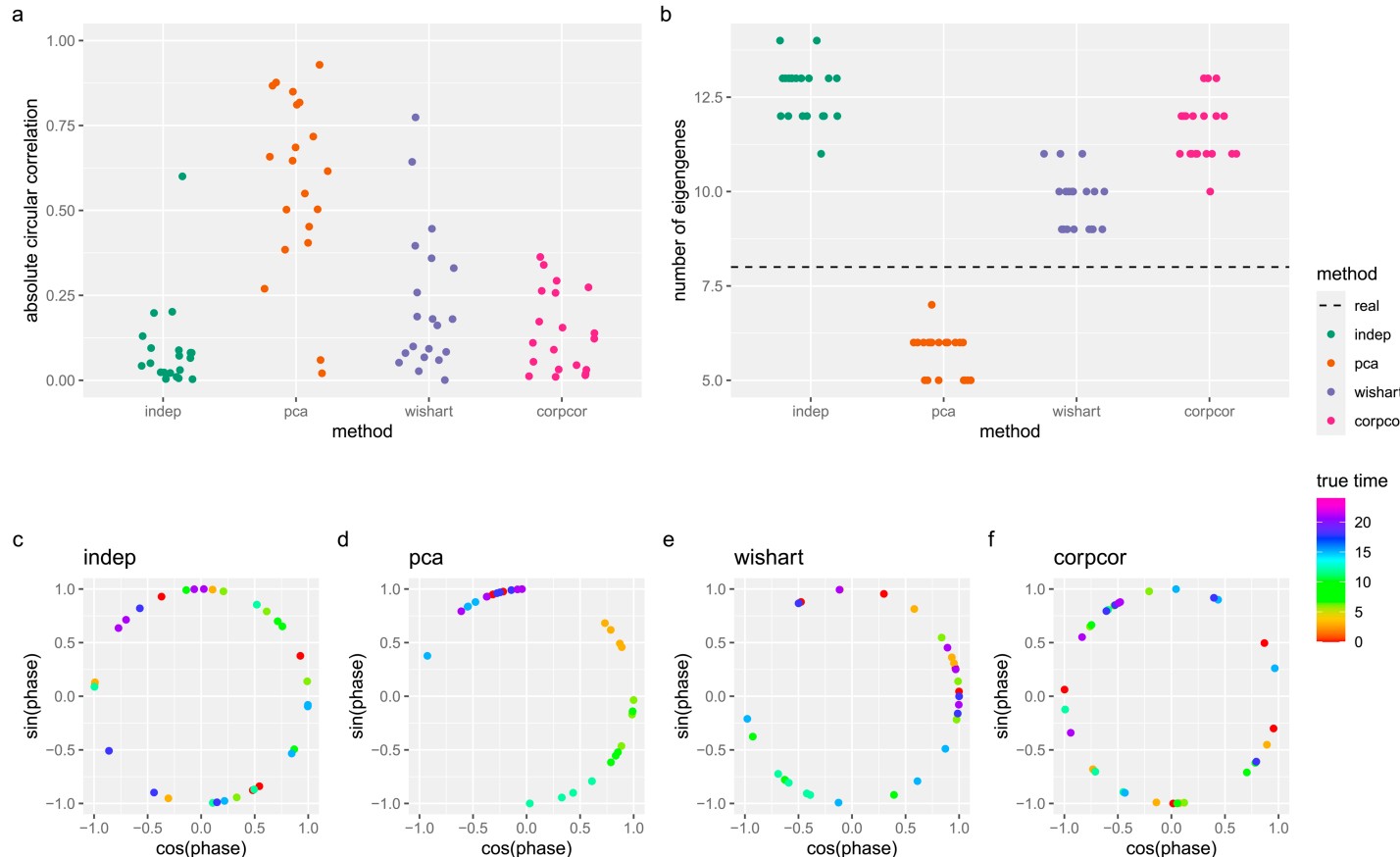

**Fig 4. Accuracy of CYCLOPS on simulated time series datasets based on mouse cortex dataset (GSE151565), using eigengenes of at least 2% variance.** (a) Absolute circular correlations between true phases and CYCLOPS estimated phases on the simulated datasets. (b) The number of eigengenes used by CYCLOPS; the dotted line indicates the number of eigengenes used by CYCLOPS on the real data (5). (c-f) Examples of CYCLOPS estimated phases on the simulated datasets. CYCLOPS shows good performance when it separates out points by color (true circadian time).

work, $\Sigma$ must be symmetric and positive semi-definite, meaning that all of its eigenvalues are non-negative. This matrix is conceptually simple, since $\Sigma_{ij}$ gives the covariance of $x_i$ and $x_j$ when $x \sim N(\mu, \Sigma)$. More specifically, this is the population covariance matrix of $N(\mu, \Sigma)$, which does not generally equal the sample covariance matrix. Indeed, if $X$ is $n$ samples from $N(\mu, \Sigma)$, then $\widehat{\Sigma} := (X - \bar{X})(X - \bar{X})^T / (n-1)$ is the sample covariance matrix where $\bar{X}$ is the average of the $n$ samples. For large $n$, $\widehat{\Sigma}$ will closely approximate $\Sigma$, but we care primarily about the situation where $n$ is small. In particular, $\widehat{\Sigma}$ is at most a rank $n-1$ matrix while $\Sigma$ could be up to rank $p$. This means that $N(\mu, \widehat{\Sigma})$ and $N(\mu, \Sigma)$ are quite different distributions: every sample from $N(\mu, \hat{\Sigma})$ is contained in an $n-1$ dimensional plane. If $n$ is small, then this is very unlike real data, which typically is close to $p$ dimensional.

The difficulty then is that we only know $\widehat{\Sigma}_{\mathrm{ref}}$ from our reference dataset, but we need to choose a $\Sigma_{\mathrm{sim}}$ with which we can simulate data and the obvious choice of $\widehat{\Sigma}_{\mathrm{ref}}$ is inadequate. The most common choice is to assume $\Sigma_{\mathrm{sim}}$ is a diagonal matrix. This is the situation we want to avoid where the generated data is independent: $x_i$ and $x_j$ have zero covariance unless $i = j$. However, this has some nice properties, such as being simple and fast to simulate (just generate univariate normal data for each variable).

We describe three alternative approaches in Methods of how to choose a $\Sigma_{\text{sim}}$ that will produce data similar to the input data. All three of these rely on a specific form of $\Sigma$, namely that

$$\Sigma_{\text{sim}} = D^2 + UW^2U^T$$

where $D$ is a $p \times p$ diagonal matrix, $U$ is $p \times k$ for some $k \ll p$ and $W$ is a diagonal $k \times k$ matrix. This is a combination of an independent part (the diagonal matrix) and a low-rank part ($UW^2U^T$ is rank at most $k$).

Generating data for $\Sigma_{\text{sim}}$ of this form is highly efficient. We use two basic facts about the multivariate normal distribution. First, if $A$ is a matrix and $u \sim N(0, \Sigma)$ then $Au \sim N(0, A\Sigma A^T)$. Second, if $u \sim N(0, \Sigma_1)$ and $v \sim N(0, \Sigma_2)$ are independent, then $u + v \sim N(0, \Sigma_1 + \Sigma_2)$. Therefore, given matrices $D, U, W$ and independent random vectors $u \sim N(0, I_k)$ and $v \sim N(0, I_p)$ (where $I_k$ and $I_p$ are the $k \times k$ and $p \times p$ identity matrices), then

$$Dv + UWu \sim N(0, D^2 + UW^2U^T) = N(0, \Sigma_{\text{sim}})$$

as desired.

In contrast, when given an arbitrary $\Sigma_{\text{sim}}$, to generate a random value in $N(0, \Sigma_{\text{sim}})$, one must compute a matrix $V$ such that $VV^T = \Sigma_{\text{sim}}$. Then, using $v \sim N(0, I)$, we have $Vv \sim N(0, \Sigma_{\text{sim}})$ as desired. However, computing $V$ is done using a Cholesky decomposition or eigenvector decomposition, both of which are computationally expensive when $\Sigma_{\text{sim}}$ is large.

Lastly, we emphasize that multivariate normal distributions do not capture all, or even most, types of possible dependence. Indeed, we see this even in the 2-dimensional case where it is well known that correlation describes only a linear relationship between two variables while in reality they may have much more complex relations. In higher dimensions, the problem is only worse. So any method based off multivariate normal distributions are making large assumptions about distribution. However, it is necessary to make some assumption like this. In the next section, though, we see that "normal" part is actually not a large obstacle.

## Gaussian copula

Building on the multivariate normal distribution, a popular approach to describe dependence in a high-dimensional settings is called the Gaussian copula approach. The idea of this approach is that by applying a normalizing transform and later reversing the transformation, data that does not fit a normal distribution can still have its dependence structure described using a multivariate normal distribution. This allows the marginal (i.e., univariate) distributions of each genes to be specified separately from the dependence between genes. This operates first by transforming each gene by fitting a distribution (such as a normal distribution, Poisson, negative binomial, or other form), and then applying the fit cumulative distribution function (CDF) to the observed values. Finally, those are fed to a standard normal distribution's inverse CDF to obtain values that are approximately normally distributed. These values are then used to compute a covariance matrix $\Sigma_{\text{sim}}$ and the data is assumed to follow a multivariate normal distribution in $p$ dimensions with that covariance matrix.

Here, we describe the approach using the form of covariance matrix $\Sigma_{\text{sim}} = D^2 + UW^2U^T$ as above. Once data is obtained $Z \sim N(0, \Sigma_{\text{sim}})$, then one can undo the normalizing transformation to obtain data with the same marginal distributions as the fit marginal distributions but with dependence determined by $\Sigma_{\text{sim}}$. We describe this in detail:

1. Fit marginal distributions to each feature in $X$ to determine CDFs $F_i$ for each feature.
2. Apply normalizing transform to $X$ by setting $Z_{ij} = \Phi^{-1}(F_i(X_{ij}))$ where $\Phi$ is the CDF of the standard normal distribution.
3. Compute $D$, $U$, $W$ matrices from $X$ by one of three methods (described later).
4. Generate $k$ i.i.d. standard normally distributed values $u$ and $p$ i.i.d standard normally distributed values $v$.
5. Set $z' = UWu + Dv$.
6. Output the vector $x'$ where $x'_i = F_i^{-1}(\Phi(z'))$.

The generated data vector $z'$ has covariance matrix $\Sigma_{\text{sim}} = D^2 + UW^2U^T$. Moreover, we require that $\Sigma$ satisfies that $(\Sigma_{\text{sim}})_{ii}$ is approximately 1 for each $i$. That guarantees that the output $x'$ has each entry with the same marginal distributions $F_i$ as was originally fit and inherits gene-gene dependence from $Z'$. This method is computationally efficient, taking hardly any more time or memory than simulations without dependence.

Below, we describe the three methods for selecting the components $D, U, W$ of the covariance matrix.

## PCA method

The first of our three methods attempts to match the top $k$ PCA components of $Z$, the reference data set after applying the normalizing transform. Specifically, let $u_1, \ldots, u_k$ be the left singular vectors of $Z$ with $\lambda_1, \ldots, \lambda_k$ the corresponding top $k$ singular values and let $U$ be the $p \times k$ matrix with columns $u_i$. This method computes $\Sigma_{\text{sim}}$ such that $u_i^T \Sigma_{\text{sim}} u_i = \lambda_i^2$, i.e. that the variance in the direction of $u_i$ exactly matches the reference dataset's variance in that same direction. One solution is to use the sample covariance matrix, but that is not full rank and would match for all $i \leq n$ instead of just $i \leq k$. Instead, we use the following:

1. Compute $A_{ij} = \delta_{ij} - \sum_\ell U_{\ell,i}^2 U_{\ell,j}^2$ and $B_i = \lambda_i^2/(n-1)^2 - \sum_\ell U_{\ell,i}^2 (\widehat{\Sigma}_{\text{ref}})_{\ell\ell}$ where $\delta_{ij}$ is the Kronecker delta and $\widehat{\Sigma}_{\text{ref}} = ZZ^T/(n-1)$ is the covariance matrix of $Z$.
2. Solve $Aw = B$ and set $W$ to be the diagonal matrix with $w$ along its diagonal.
3. Set $U$ to be the $p \times k$ matrix with columns $u_i$.
4. Set $D$ to be the diagonal matrix with $D_{ii}^2 = (\widehat{\Sigma}_{\text{ref}})_{ii} - (UW^2U^T)_{ii}$, which is the remaining variance.

Steps 1 and 2 give that $u_i \Sigma_{\text{sim}} u_i^T = \lambda_i$ for $i = 1, \ldots, k$. Step 4 ensures that $(\Sigma_{\text{sim}})_{jj} = 1$ for $j = 1, \ldots, p$.

## Spiked Wishart method

The second method also makes use of PCA but has a different objective. If $n$ samples are drawn from $N(0, \Sigma_{\text{sim}})$ then we want the variances of their PCA components to match those of the reference dataset. Specifically, let $\lambda_1, \ldots, \lambda_{n-1}$ be the $n-1$ non-zero singular values of $Z$ ($\lambda_n$ is always approximately zero due to the normalization procedure), and let $\lambda'_1, \ldots, \lambda'_{n-1}$ be the singular values of $Z'$ where $Z'$ has $n-1$ columns each iid $N(0, \Sigma_{\text{sim}})$. Then we want to choose $\Sigma_{\text{sim}}$ such that $E[\lambda'_i] = \lambda_i$ for each $i$, where $E[Y]$ denotes the expectation of the random variable $Y$.

Since the distribution of the $\lambda'_i$ does not have a known analytic solution, we approximate this situation with the spiked Wishart distribution. The rank $k-1$ Wishart distribution is the distribution of sample covariance matrices of $Z$ whose $n$ columns of $Z$ are iid $N(0, \Sigma_{\text{sim}})$.

The spiked Wishart is the special case where $\Sigma_{sim}$ has $k$ arbitrary eigenvalues and the remaining are all equal to a constant. Note that the singular values of $Z$ are the square roots of the eigenvalues of $\widehat{\Sigma}_{ref}$. While our case has $\Sigma_{sim}$ non-diagonal, $\Sigma_{sim}$ may be diagonalized by orthogonal rotations due the spectral theorem, and orthogonal rotations do not change the singular values of $Z$. Therefore, the distribution of singular values is not affected by the assumption that $\Sigma_{sim}$ is diagonal. Moreover, for the form $\Sigma_{sim} = D^2 + UW^2U^T$ where $p$ is very large, each column of $U$ is typically very close to orthogonal to any standard basis vector. Therefore, when $D = cI$ for some constant $c$, we can approximate $\Sigma_{sim}$ as having $k$ arbitrary eigenvalues from $W^2$ and $n$ remaining eigenvalues all equal to $c^2$. This is a spiked Wishart distribution.

However, the spiked Wishart distribution also has no known analytic solution for the distribution of its eigenvalues either. Therefore, we use an efficient sampling and stochastic gradient descent method that we recently described [27]. Since the normalizing transform has been applied, $c$ will be close to one and $(\Sigma_{sim})_{ii} \approx 1$ for each $i$.

Specifically, we do:

1. Set $U$ to be the $p \times k$ matrix with columns $u_i$, the left singular vectors of $Z$.
2. Compute $w_1, \ldots, w_k$ and $c$ by stochastic gradient descent minimizing $\sum_i (E[\lambda_i'] - \lambda_i)^2$ for $\Sigma_{sim}$ diagonal with entries $w_1^2, \ldots, w_k^2, c^2, \ldots, c^2$ [27].
3. Set $W$ diagonal with the entries $w_1, \ldots, w_k$.
4. Set $D = cI$.

## Corpcor method

The `corpcor` package [4,5] computes a James-Stein type shrinkage estimator for the covariance matrix. For large $p$, this greatly improves the estimate of the covariance matrix by introducing a little bias towards zero correlations and equal variances of genes. It computes optimal values of $\lambda_1$, and $\lambda_2$, its two regularization coefficients. It then uses $\lambda_1$ to linearly interpolate the sample covariance matrix towards the identity matrix $I$ and $\lambda_2$ to interpolate the vector of variances towards the median variance value. Since the sample covariance matrix is rank at most $n$, we again obtain a matrix of the form $\Sigma_{sim} = D^2 + UW^2U^T$.

This algorithm is:

1. Compute the $\lambda_1$ and $\lambda_2$ values from `corpcor::estimate.lambda` and `corpcor::estimate.lambda.var` functions on $Z$, respectively.
2. Set $D$ to be diagonal with $D_{ii}^2 = \lambda_1(\lambda_2\sigma_{med} + (1-\lambda_2)\sigma_i)$ where $\sigma_i$ is the standard deviation of the $Z_{i.}$ and $\sigma_{med}$ is the median of the $\sigma_i$.
3. Set $U$ to be $\sqrt{1-\lambda_1}SZ/\sqrt{n-1}$ where $S$ is the diagonal matrix with $S_{ii}^2 = \lambda_2\sigma_{med}/\sigma_i + (1-\lambda_2)$.
4. Set $W$ to the identity.

## Discussion

Here we have described the well-known Gaussian copula approach and argued that a specific form of covariance matrix is well tailored to omics data simulation. We developed three methods using this form of covariance matrix which can be used to mimic a reference dataset for simulation. All of these methods use a multivariate normal distribution as an intermediate step and therefore substantially restrict the kinds of dependence that can be simulated. However, when operating in a high-dimensional space some simplification is required.

To encourage adoption of dependence in simulated omics data, we developed `dependentsimr`, an R package that generates omics-scale data with realistic correlation. This implementation is efficient and simple, requiring just two lines of code to fit a model to a reference dataset and then simulate data from it. We demonstrated this package on RNA-seq data, using the DESeq2 method to fit negative binomial marginal distributions, as well as a metabolomics dataset using normal marginal distributions. Our package is even more general and supports normal, Poisson, negative binomial, and arbitrary ordered discrete distributions using the empirical CDF. Moreover, it can support multi-modal data such as is increasingly common in multi-omics. In addition, the techniques described here are relatively simple to implement and we hope they can see broader adoption outside of our package.

We demonstrated that our methods are highly efficient and easily scale to the large feature counts typical of omics datasets. While runtime of simulators is typically not critical, we posit that it introduces a sufficient inconvenience to have slowed the adoption of dependence in omics simulation and could be particularly cumbersome whenever large numbers of simulated datasets are needed. Moreover, high memory use could be an even larger barrier than run-time, and is also dramatically improved by our methods. Of note is that alternative methods are typically run with only a subset of genes chosen, for example, 500-2000 genes for scDesign2 [14], 1000 genes for scDesign3 [16], and 621 for SeqNet [17]; presumably these limitations are due to the high resource requirements. In contrast, our methods can operate on the entire space of genes, avoiding an arbitrary choice of cutoff, though it should be noted that correlation is most meaningful in high-expressed, high-variance genes and so subsetting genes is reasonable.

We demonstrated the importance of including gene-gene dependence in simulated data by two benchmarking applications. In the first, DESeq2 results were substantially more variable when simulating with gene-gene dependence. This indicates that DESeq2 had an increased chance of a larger simultaneous number of false positives, as well as increased chance of fewer false positives, when data included realistic dependence. We therefore recommend that benchmarking of differential expression methods include gene-gene dependence whenever possible. This is despite the fact that differential expression is largely addressing a problem of the behavior of individual genes. We note that DESeq2's observed anti-conservative nature is most present at very low FDR thresholds and was typically associated with only a single digit number of false positive genes. Moreover, more modern methods to control FDR are compatible with DESeq2 and have been shown to be appropriately conservative [28].

In the second application, CYCLOPS performance in estimating circadian phases was sensitive to dependence structure of the data under some configurations. Unlike DESeq2, CYCLOPS explicitly makes use of the correlations of genes and therefore this result is not surprising but still demonstrates the potential impact of the assumption of independence when benchmarking.

Our comparisons to a real dataset show that none of our three methods are able to exactly capture all aspects of the real dataset. In particular, the gene-gene correlations were too high in the PCA method and too low in the spiked Wishart and corpcor methods. Surprisingly, the spiked Wishart and corpcor methods improved in this metric only slightly compared to the simulations with independent genes. These observations demonstrate that there is room for future improvements over independent data in these techniques, possibly incorporating more recent developments in copulae [29]. This could demonstrate the limitations of methods based on the multivariate normal distribution or of the low-rank approximation used by all three proposed methods. Nonetheless, these methods represent significant improvements by other metrics and we recommend the inclusion of some dependence in nearly every simulated omics dataset.

We observe that our PCA method generally overshoots the level of correlation in the real data set. The corpcor method typically undershoots. The spiked Wishart method also undershoots but less so than the corpcor method and therefore we believe represents the best starting point for users. However, we emphasize that in any application, the users should compare their simulated data to the reference dataset and this can be used to determine the best method for the specific application at hand, which could vary depending on data set.

We compared our simulated data to that of the R package SPsimSeq [13], due to its popularity, support for bulk RNA-Seq, and use of Gaussian copulas. In contrast to our package, SPsimSeq uses a two-step randomization. In short, it first fits distributions to each genes and then bins those distributions into discrete buckets. Then, new data is generated by drawing from a multivariate normal distribution whose covariance matrix equals the sample covariance matrix of the reference dataset. These values are used to choose which bucket each gene is drawn from. Then, each gene's final value is drawn uniformly and independently from within its chosen bucket. Therefore, SPsimSeq induces correlation in the first multivariate normal draw, but then injects additional independence in the second uniform step. This two-step process makes it challenging to compare on theoretical grounds to our proposed methods since it will depended upon parameters such as the number of buckets used (more fine-grained buckets will produce higher correlation). SPsimSeq is more specialized and full-featured for RNA-seq simulation, providing, for example, native differential expression (DE) options. In comparison, our dependentsimr package requires manually setting marginal expression values to inject DE, but also supports other marginal distributions for situations outside of RNA-seq.

Other Gaussian copula-based R packages include `copula`, `bindata`, `GenOrd`, and `SimMultiCorrData`, the last of these being the most comprehensive. The `bigsimr` package provides faster implementations of these methods which scale up to omics-level data. However, even this is computationally demanding; their paper references generating 20,000-dimensional vectors in "under an hour" using 16 threads. All of these packages provide more flexibility in specifying dependence and therefore the longer run-times may be unavoidable for use cases where researchers need to parameterize the dependence structure. Another package that optimizes the speed of random multivariate normal vector generation is the `mvnfast` package for R [30], however, we show that this does not provide a meaningful speedup for our use-cases, likely due to the reliance on the Cholesky decomposition, see Fig 1C.

While we describe one special form of $\Sigma_{\text{sim}}$ where generation is highly efficient, there exist many other possible covariance matrix forms which are promising directions for future work. Block diagonal covariance matrices are one such example which can efficiently represent even high-dimensional data [31] and lend themselves to efficient simulation. Other approaches include sparse matrix methods [32]; Bickel and Levina 2008 [33]; [6] which yield exactly zero correlation between some variables, but it is less clear how to simulate data from these efficiently. Iterative sampling methods [34] are one approach that could exploit the sparse nature and may even extend to sparse inverse covariance methods [7].

## Supporting information

**S1 Fig. Evaluation of our simulation on a metabolomics dataset.**
(PDF)

**S2 Fig. Evaluation of our simulation on a fly whole-body RNA-seq dataset.**
(PDF)

**S3 Fig. Evaluation of our simulation on a mouse cortex RNA-seq dataset.**
(PDF)

**S1 Text. Supplemental methods.**
(PDF)

## Author contributions

**Conceptualization:** Gregory R. Grant.

**Data curation:** Jianing Yang.

**Formal analysis:** Jianing Yang, Thomas G. Brooks.

**Methodology:** Thomas G. Brooks.

**Project administration:** Gregory R. Grant.

**Software:** Jianing Yang, Thomas G. Brooks.

**Supervision:** Gregory R. Grant.

**Visualization:** Jianing Yang, Thomas G. Brooks.

**Writing – original draft:** Jianing Yang, Thomas G. Brooks.

**Writing – review & editing:** Gregory R. Grant.

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
