## [Decision Letter · Decision Letter 0]

17 Apr 2025

PCOMPBIOL-D-25-00243

Generating Correlated Data for Omics Simulation

PLOS Computational Biology

Dear Dr. Brooks,

Thank you for submitting your manuscript to PLOS Computational Biology. After careful consideration, we feel that it has merit but does not fully meet PLOS Computational Biology's publication criteria as it currently stands. Therefore, we invite you to submit a revised version of the manuscript that addresses the points raised during the review process.

Please submit your revised manuscript within 60 days Jun 17 2025 11:59PM. If you will need more time than this to complete your revisions, please reply to this message or contact the journal office at ploscompbiol@plos.org. Please include the following items when submitting your revised manuscript:

We look forward to receiving your revised manuscript.

Kind regards,

Jie Liu

Academic Editor

PLOS Computational Biology

Jian Ma

Section Editor

PLOS Computational Biology

**Journal Requirements:**

3) Your manuscript's sections are not in the correct order.  Please amend to the following order: Abstract, Introduction, Results, Discussion, and Methods

5) Thank you for stating "All data used is available with accession numbers GSE151923,GSE81142, GSE151565." Please include the repository name in your Data Availability Statement.

6) Please include the grant recipients in the Funding Information tab.

7) Thank you for stating "Source code for all simulations and figures in this plot is available at github.com/itmat/dependent_sim_paper/." This link reaches a 404 error page. Please amend this to a working link.

**Reviewers' comments:**

Reviewer's Responses to Questions

**Comments to the Authors:**

**Please note that two reviews are uploaded as attachments.**

Reviewer #1: Please see attached.

Reviewer #2: This study presents three methods for simulating correlated omics data using a Gaussian copula approach with a covariance matrix decomposition. The authors demonstrate their utility in benchmarking computational tools such as DESeq2 and CYCLOPS. This paper is quite interesting. However, there are some major and minor issues need to be addressed:

Major issues:

1. The manuscript describes a data simulation approach. Could you provide a clear rationale for this specific goal? Also, the authors demonstrate the covariance matrix decomposition into diagonal and low-rank components. Can you show what the hypothesis behind it. More specifically, how does this decomposition compare to other methods of modeling correlation structures in RNA-seq data?

2. For the introduction section, could you include more related works on probabilistic models and generative models that are relevant to data simulation and benchmarking? Providing a broader context on these approaches would strengthen the foundation of the study.

3. While the study compares simulated data to real datasets, the validation metrics primarily focus on variance and correlation structure. Could you provide additional validation using biological benchmarks (e.g., differential analysis, functional enrichment) could further support the biological relevance of the simulated datasets.

4. The results indicate increased variance in DESeq2 outcomes when using correlated data, but the practical implications for differential expression analysis are not fully explored. What does this increased variance represent?

5. By comparing with the real datasets, the authors compared the independent genes with the dependent genes with various methods. But none of the existing simulation approach has been compared. Can you benchmark with existing methods and show your simulation performances, e.g scDesign, ZINB-WAVE, etc.?

6. What is your conclusion for this three different methods? It seems like there is no consistent best methods across different datasets.

Reviewer #3: Main review comments are uploaded as an attachment.

Overall, this article has a lot of writing problems and the results are not particularly convincing.

**Have the authors made all data and (if applicable) computational code underlying the findings in their manuscript fully available?**

Reviewer #1: Yes

Reviewer #2: Yes

Reviewer #3: **No: **The author states that “Source code for all simulations and figures in this plot is available at github.com/itmat/dependent_sim_paper/”. However, currently there is no content in the provided link. The author should make sure the source code is available at the provided link. The provided github repository for the R package seems to be valid.

PLOS authors have the option to publish the peer review history of their article (what does this mean?). If published, this will include your full peer review and any attached files.

Reviewer #1: No

Reviewer #2: No

Reviewer #3: No

**Figure resubmission:**
---

## [Decision Letter · Decision Letter 1]

4 Aug 2025

Dear Dr Brooks,

We are pleased to inform you that your manuscript 'Generating Correlated Data for Omics Simulation' has been provisionally accepted for publication in PLOS Computational Biology.

Best regards,

Jie Liu

Academic Editor

PLOS Computational Biology

Jian Ma

Section Editor

PLOS Computational Biology

There are still a few minor revision suggestions. That will be great if the authors can make the changes eventually.

Reviewer's Responses to Questions

**Comments to the Authors:**

Reviewer #1: This revision has resolved my main concerns. Some remaining points for clarification:

1. The setup described in response to my minor point (2) makes sense. I would still considering calling the samples "conditionally independent" or "independent conditional on time" rather than independent, sinec that might give the impression of the samples being independent without any conditioning at all.

2. The additional comparison with SPSimSeq is worthwhile. There are many simulators that could have been used, and the manuscript should justify why SPSimSeq was chosen. My understanding is that it was chosen because it is also based on Gaussian copulas, but this wasn't entirely clear.

3. The new runtime benchmarking is useful but potentially raises questions about how this proposal compares with the relatively large literature on fast simulation of multivariate gaussians (e.g., mvnfast). While it is helpful to mention bigsimr, it does not seem fair to dismiss the entire literature based on this one package's performance, and a few extra references could help prevent any risk of misinterpretation.

Reviewer #2: I have read the Authors' response to the reviewers and am satisified that they have addressed all of my questions and concerns.

Reviewer #3: It seems that the authors have addressed most of my review comments. I have the following four follow-up comments:

1. When responding to review comment #8, the authors included some theoretical details on how the random sampling works.

(1) “We use two basic facts about the multivariate distribution.” Here it should be “multivariate normal distribution”, not “multivariate distribution”.

(2) “if u ~ N(0, Sigma_1) and v ~ N(0, Sigma_2), then u + v ~ N(0, Sigma_1 + Sigma_2).” This only holds when u and v are independent or at least uncorrelated. The authors need to explicitly state this assumption to avoid confusion, and they also need to explain why this independence or uncorrelated assumption applies here.

2. The authors may consider spending some more efforts on revising the language of the paper. Right now many places read unsmooth to me, but if I read it a few more times, I could get the main point.

3. In the Data Availability Section, the link to the Metabolomics data from MetaboAnalyst does not work when I click it.

4. I found a typo: Row 6 of the caption of Figure 1: should be corpcor, not coprcor.

Reviewer #4: This manuscript presents a set of efficient methods for simulating high-dimensional omics data with realistic correlation structures using a Gaussian copula framework, addressing a key limitation in current simulation tools that often assume independence across features. The authors propose three covariance matrix strategies (PCA, spiked Wishart, and corpcor), implement them in the dependentsimr R package, and demonstrate their utility through benchmarking applications with DESeq2 and CYCLOPS. The revised version thoughtfully and thoroughly addresses prior reviewer concerns. The manuscript is substantially improved. As a minor suggestion, it would be helpful to provide a brief roadmap for extending the framework to additional omics modalities (e.g., single-cell, proteomics, or multi-omics), which would strengthen the paper’s generalizability and long-term impact.

**Have the authors made all data and (if applicable) computational code underlying the findings in their manuscript fully available?**

Reviewer #1: Yes

Reviewer #2: Yes

Reviewer #3: Yes

Reviewer #4: None

PLOS authors have the option to publish the peer review history of their article (what does this mean?). If published, this will include your full peer review and any attached files.

Reviewer #1: **Yes: **Kris Sankaran

Reviewer #2: No

Reviewer #3: No

Reviewer #4: No

---

## [Editor Report · Acceptance letter]

PCOMPBIOL-D-25-00243R1

Generating Correlated Data for Omics Simulation

Dear Dr Brooks,

I am pleased to inform you that your manuscript has been formally accepted for publication in PLOS Computational Biology. Your manuscript is now with our production department and you will be notified of the publication date in due course.

With kind regards,

Livia Horvath
